# An Imine-Based Porous 3D Covalent Organic Polymer as a New Sorbent for the Solid-Phase Extraction of Amphenicols from Water Sample

**DOI:** 10.3390/molecules28083301

**Published:** 2023-04-07

**Authors:** Jinjian Wei, Lengbing Chen, Rui Zhang, Yi Yu, Wenhua Ji, Zhaosheng Hou, Yuqin Chen, Zhide Zhang

**Affiliations:** 1Key Laboratory of Molecular and Nano Probes, Collaborative Innovation Center of Functionalized Probes for Chemical Imaging in Universities of Shandong, College of Chemistry, Chemical Engineering and Materials Science, Ministry of Education, Shandong Normal University, Jinan 250014, China; 2Key Laboratory for Applied Technology of Sophisticated Analytical Instruments of Shandong Province, Shandong Analysis and Test Center, Qilu University of Technology (Shandong Academy of Sciences), Jinan 250014, China

**Keywords:** imine, covalent organic polymer, solid-phase extraction, amphenicols

## Abstract

In this paper, an imine-based porous 3D covalent organic polymer (COP) was synthesized via solvothermal condensation. The structure of the 3D COP was fully characterized by Fourier transform infrared spectroscopy, scanning electron microscopy, transmission electron microscopy, and powder X-ray diffractometry, thermogravimetric analysis, and Brunauer–Emmer–Teller (BET) nitrogen adsorption. This porous 3D COP was used as a new sorbent for the solid-phase extraction (SPE) of amphenicol drugs, including chloramphenicol (CAP), thiamphenicol (TAP), and florfenicol (FF) in aqueous solution. Factors were investigated for their effects on the SPE efficiency, including the types and volume of eluent, washing speed, pH, and salinity of water. Under the optimized conditions, this method gave a wide linear range (0.1–200 ng/mL) with a high correlation coefficient value (R^2^ > 0.99), low limits of detection (LODs, 0.01–0.03 ng/mL), and low limits of quantification (LOQs, 0.04–0.10 ng/mL). The recoveries ranged from 83.98% to 110.7% with RSDs ≤ 7.02%. The good enrichment performance for this porous 3D COP might contribute to the hydrophobic and π–π interactions, the size-matching effect, hydrogen bonding, and the good chemical stability of 3D COP. This 3D COP-SPE method provides a promising approach to selectively extract trace amounts of CAP, TAP, and FF in environmental water samples in ng quantities.

## 1. Introduction

Amphenicols, including chloramphenicol (CAP), thiamphenicol (TAP), and florfenicol (FF), are bacteriostatic antibiotics with broad-spectrum activity [1]. Due to their high efficacies and good bacterial inhibition, they are widely used in animal aquaculture and animal husbandry to cure and control infectious diseases. However, CAP, TAP, and FF have various severe side effects in humans such as leukemia, aplastic anemia, and gray baby syndrome [2,3]. This suggests that antibiotic residues in the food chain can potentially be harmful to human health. As a result, many countries have strictly regulated or banned the usage of CAP, TAP, and FF in animals to prevent them from entering the food chain [4,5,6]. The maximum residue limits of CAP, TAP, the total quantity of FF and its major metabolite florfenicol amine (FFA) are regulated to 0.3 μg kg^−1^, 50 μg kg^−1^, and 100 μg kg^−1^, respectively, in poultry tissues by EU and China. However, monitoring the trace of residue amphenicols in the environment to protect humans’ health and safety remains an enormous challenge because of the limited and efficient analytical protocols. Therefore, it is important to establish a simple, sensitive, and accurate method to detect residual CAP, TAP, and FF in vivo and in vitro.

For this, several analytical methods, such as subcritical water extraction (SWE) [7], liquid chromatography/tandem mass spectrometry (LC-MS/MS) [8,9], gas chromatography-mass spectrometry (GC-MS) [10,11], enzyme-linked immunosorbent assay (ELISA) [12,13], and immunochemical techniques [14], have been developed to detect amphenicols in fish [15,16], urine [17], chicken [18], honey [17,19,20,21], milk [17,20], and shrimp [10]. However, a limited number of methods to detect trace amounts of CAP, TAP, and FF in water samples were reported [22,23,24]. The fundamental research to detect CAP, TAP, and FF in water can be used to extend analytical methodologies for real environmental water samples. Solid-phase extraction (SPE) is a classic extraction technique that has some merits over other extraction protocols such as a stable extraction efficiency, short extraction time, high enrichment coefficient, and low solvent usage [25]. During SPE, the selection of a suitable sorbent is pivotal to obtain a good enrichment and high recovery results [26]. Previously, some sorbents such as molecular-imprinted polymers (MIPs) [27,28], carbon nanomaterials [29,30], metallic nanomaterials [31,32], and metal-organic frameworks (MOFs) [33,34] have been investigated. Even some materials, as mentioned above for SPE, were reported; the development and characterization of novel sorbents with good adsorption capabilities, high selectivity, and excellent chemical and thermal stabilities are still hot topics in SPE research [26].

In contrast to MOF materials with metal counterparts, light elements constituting porous organic polymers (POPs) have attracted broad interest in the past years on account of their wide applications [35,36]. The surface area and porosity of POPs could be well controlled by the polymerization between the specifically designed monomers. The POPs were generally classified into amorphous structures such as porous aromatic frameworks [37,38], conjugated microporous polymers [39,40], hyper-crosslinked polymers [41,42], and crystalline covalent organic frameworks (COFs) [43,44,45]. Among these, COFs showed various merits such as high crystallinity, highly-ordered mesoporous structures, low density, high surface area, and easy functionalization [43]. Up to now, various two-dimensional (2D) and some three-dimensional (3D) COFs constructed by various building blocks and functional groups have been reported [46,47,48,49]. However, it is generally difficult to efficiently control the crystallization process and high experimental techniques are needed, thus hindering their practical applications. On the other hand, covalent organic polymers (COPs) also exhibited COF-like characteristics such as controllable pore size, highly crosslinked porous structure, and functionality [50]. In contrast to COFs, COPs generally have an amorphous to semi-crystalline nature, and their fabrication procedures are simpler than those of COFs [51]. Therefore, COPs showed widespread applications in separation [51], therapy [52], and catalysis [53]. However, previous reports have been mainly based on 2D COPs [51,52,53,54], while only a limited number of studies have explored the construction of 3D COPs [55,56]. Few applications of 3D COPs as new sorbents for the SPE of amphenicol drugs in water have been reported.

In this paper, we report the construction of an imine-based porous 3D COP, which was used for the SPE and enrichment of CAP, TAP, and FF in water in ng quantities. Different factors including the type and volume of eluent, washing speed, pH, and salinity of water were investigated and optimized. Furthermore, the working curve equation, linear range, correlation coefficient, limit of detection, and quantification under the optimal conditions were obtained. The possible enrichment mechanism was initially discussed according to the structural relationship between analytes and 3D COPs. The reusability of the 3D COP was finally evaluated.

## 2. Results and Discussion

### 2.1. Synthesis and Characterization of Imine-Based 3D COP

An imine-based porous 3D covalent organic polymer (COP) was synthesized via solvothermal condensation (Figure 1). Tetrakis(4-formylphenyl)-methane (TFPM) and 4,4′-diaminobiphenyl (DABP) were suspended in a mixture of 1,4-dioxane and mesitylene in the existence of acetic acid with subsequently heating at 120 °C for 3 days. The Fourier-transform infrared (FT IR) spectra of the synthesized polymer (Figure 1a) show the almost disappearance of the –N–H stretching vibration from 3100 to 3400 cm^−1^ and the appearance of a new stretching vibration band at 1619 cm^−1^ for a –C=N bond, indicating the occurrence of a condensation reaction. The stretching signal at 1695 cm^−1^ corresponded to the unreacted residue –CHO group at the edges/termini of the 3D COP. Powder X-ray diffraction (PXRD) data showed that the peak at 7.12° of the PXRD patterns (Appendix A) was not corresponding to the Bragg peaks (5.19°) of space group I41/a for the earlier reported COF [48]. In addition, other reflection peaks at 8.49, 12.15, and 13.46° were not observed in the present PXRD data. Furthermore, the intensity of the reflection peak at 7.12° was weak, indicating that the synthesized polymer was not well crystalized, at least not the same 3D COF as previous report [48]. Thermogravimetric analysis (TGA) data (Figure 1b) showed that the 3D COP was thermally stable below 400 °C, revealing the good thermal stability of the 3D COP. N_2_ adsorption–desorption isotherms were obtained at 77 K to evaluate the porosity of 3D COPs. Brunauer–Emmer–Teller (BET) nitrogen adsorption data (Figure 1c) displayed a sharp gas uptake at low pressure (<0.1 P/P_0_), indicating that the 3D COP exhibited a microporous structure. The literature revealed that the inclination of isotherms in the 0.8–1.0 P/P_0_ range showed the existence of textural mesopores due to the agglomeration of polymers [48]. According to the data in Figure 1c, the calculated BET surface area (169 m^2^/g) is smaller than that (653 m^2^/g) of the reported COF material constructed by the same monomers [48]. The poor crystallinity of 3D COPs might induce the lower experimental N_2_ adsorption or calculated surface area than those of the well-crystalized COF. According to the size distribution in Figure 1d, the calculated average pore width was 1.03 nm. Based on the above data, this polymer was defined as a porous 3D COP. Scanning electron microscopy (SEM) (Figure 1e) and transmission electron microscopy (TEM) (Figure 1f) revealed that the 3D COP exhibited rod-like aggregations.

### 2.2. Optimization of SPE

For the SPE process, the types and volume of eluent, washing speed, pH, and salinity of water significantly influenced the extraction efficiency. As described in a previous report [57], the optimized parameters were applied to optimize subsequent parameters. To ensure data reproducibility, every optimization step was repeated three times.

#### 2.2.1. Effect of Eluent

The purpose of elution is to flush the analytes adsorbed on the solid phase as much as possible without destroying the original properties of the filler. As seen in Figure 2a, six solvent system, including acetonitrile, acetone, methanol, methanol/acetic acid (V_methanol_/V_acetic acid_, 99:1), methanol/formic acid (V_methanol_/V_formic acid_, 97:3), and methanol/trifluoroacetic acid (V_methanol_/V_trifluoroacetic acid_, 95:5) could desorb the three analytes from the solid phase. For the single solvent system, methanol provided a better recovery compared with acetonitrile and acetone. The addition of formic acid in methanol (V_methanol_/V_formic acid_, 97:3) obviously improved the recovery of FF, TAP, and CAP, among which the recovery of FF reached 80%. Therefore, the methanol/formic acid solution (V_methanol_/V_formic acid_, 97:3) was used to evaluate the influence of elution volume on recovery. From the data in Figure 2b, when the volume of elution increased from 5 to 15 mL, the recovery of the three analytes increased to 90%. Further increasing the volume of elution to 20 mL did not obviously change the recovery; therefore, an elution volume of 15 mL was chosen for subsequent investigations.

#### 2.2.2. Effect of Flow Velocity and pH of Loading Sample Solution

The flow velocity of sample solutions was related to the extraction equilibrium and kinetics; thus, it directly influenced the adsorption speed of target molecules on the porous 3D COP. The data in Figure 3a clearly show that the recovery of three analytes decreased dramatically by reaching the higher flow speed (larger than 0.5 mL/min). On the other hand, when the speed was lower than 0.5 mL/min, a slight change in the recovery was observed, indicating that extraction equilibrium was achieved. Thus, 0.5 mL/min was chosen as the optimized flow velocity.

The pH of the water sample may influence the charge state of analytes in solution, which can further influence the extraction efficiency. Figure 3b showed that the recovery of CAP, TAP, and FF was relatively high when the pH of water ranged from 3–6. However, as the pH increased from 7 to 10 upon the addition of NaOH solution, the recovery obviously decreased. This was probably caused by increasing the solubility of analytes in water through protonation under acidic conditions, which weakened the hydrophobic interactions between the analytes and 3D COP. Thus, a pH of 6 was chosen as the optimal condition for the following investigations.

#### 2.2.3. Effect of Salinity of Water Sample

Previous reports revealed that the addition of salt in water samples increases the ion concentration and ionic strength of a solution, which positively influences the adsorption efficiency of hydrophilic compounds. As shown in Figure 4, when the salinity increased from 0 to 15, the recovery of the three analytes increased. As the salinity increased beyond 15, the recovery decreased dramatically. This was probably due to the competitive adsorption of excess Na^+^ or Cl^−^ on the surface or pores of the 3D COP, which exceeded the salt-out effect according to a previous investigation [58]. Hence, a salinity of 15 was chosen for further investigations.

Briefly, the effects of the five factors—eluent type, eluent volume, loading flow rate, pH, and salinity—on the extraction process were optimized and listed as follows: a methanol formic acid solution (V_methanol_/V_formic acid_, 97:3); an eluent volume of 15 mL; a loading flow rate, 0.5 mL min^−1^; a water sample pH of 6; and a salinity of 15.

### 2.3. Methodological Investigation

Under the optimized conditions mentioned above, method validation including the linear range, correlation coefficient (R^2^), limits of detection (LODs), limits of quantification (LOQs), precision, and relative standard deviation (RSD) are listed in Table 1. CAP, TAP, and FF were added to water samples to increase the analyte concentrations to 0.1–200 ng/mL for CAP and those of FF and TAP were 0.3–200 ng/mL. The LODs obtained from three replicates were 0.01 ng/mL for CAP, and 0.03 ng/mL for FF and TAP, respectively, with a signal-to-noise ratio (S/N) of 3. The LOQs were 0.04 ng/mL for CAP and 0.10 ng/mL for FF and TAP, with a baseline S/N of 10. This indicated the high sensitivity of this method and demonstrated that it was sufficiently sensitive to detect trace amounts of CAP, TAP, and FF in real water samples. Furthermore, the linear correlation coefficients (R^2^) of working curves for CAP were larger than 0.99, while R^2^ of FF and TAP was larger than 0.999. This demonstrated that the concentration of substances in water samples showed excellent linear relationship with the peak area when the concentration of CAP was 0.1–200 ng/mL and when FF and TAP ranged from 0.3–200 ng/mL. As shown in Table 1, the relative standard deviation for intra-day and inter-day precisions ranged from 1.72 to 3.15% and 3.70 to 4.87% respectively. The RSDs (*n* = 3) of the extraction performance ranged from 3.6% to 5.3%, indicating the good reproducibility of the 3D COP-SPE method. These data demonstrate that the 3D COP-SPE method was accurate and could reliably and reproducibly extract CAP, FF, and FF from water samples.

The standard recovery was further investigated to evaluate the precision and accuracy of the method. Table 2 and Appendix A list the recoveries and RSD%, which demonstrate that the 3D COP-SPE method can be applied to determinate CAP, TAP, and FF in water matrices with good precision and accuracy.

### 2.4. Reusability of 3D COP

The extraction recovery over 10 continuous cycles was carried out to evaluate the reusability of the 3D COP. From the data in Figure 5a, the recovery showed almost no significant change as the number of extraction cycles increased, demonstrating that the 3D COP used in this research has good recycling performance. A comparison of the FTIR spectra (Figure 5b) after 100 cycles under different conditions further indicates the good stability of the 3D COP after extraction. The corresponding calculated BET surface area was 120 m^2^/g, slightly lower than that (169 m^2^/g) before extraction. However, this change of surface area did not obviously affect the extraction efficiency.

### 2.5. Proposed Enrichment Mechanism

From the structural perspective of CAP, TAP and FF, phenyl groups with hydrophobic properties and π donors and hydroxyl groups with hydrophilic characteristics and hydrogen bond doners are included (Figure 6). The imine-based 3D COPs with aromatic skeletons are highly hydrophobic and bearing strong π–π interaction binding sites. The above discussion shows that the imine-based 3D COP has a strong adsorption and good enrichment for each of the three analytes. The unique pore width of the 3D COP is 1.03 nm as discussed above, and the length of adsorb analytes is calculated to be 0.98 nm by Chem 3D. This indicated that the analytes could sufficiently adsorb on the pores of 3D COPs due to the size-matching effect. In addition, this process was mainly driven by the combination of the hydrophobic attraction between the hydrophobic groups of the 3D COP and the analytes and their π–π interactions. In addition to these interactions, hydrogen bonding between the lone electron pair of the imine and the hydrogen of the hydroxyl group may have also contributed to the enrichment. Based on the above discussion, the good enrichment performance was ascribed to hydrophobic and π–π interactions, the size-matching effect, hydrogen bonding, and the good chemical stability of 3D COP.

### 2.6. Comparison with Previous Reports

The 3D COP-SPE method was compared with previous reports of the determination of residual amphenicol drugs in water samples. The linearity, LODs, and recoveries are summarized in Table 3. The 3D COP-SPE method exhibited a wider linearity and comparable recoveries to previously-reported methods. The LODs of 3D COP-SPE were obviously lower than those obtained from other pretreatment methods, including direct measure without treatment [24], SPME using fiber, [25] and SPE using resin [26]. Based on the above discussions, combining the 3D COP-SPE with HPLC-DAD offers a simple, accurate, and sensitive method for determining amphenicol drugs in water samples.

## 3. Materials and Methods

### 3.1. Chemicals

Standard samples of CAP, TAP, and FF were obtained from Tianjin Guangcheng Chemical Reagent Co., Ltd. (Tianjin, China). Tetrakis(4-formylphenyl)-methane was obtained from Aladdin Reagent Co., Ltd. (Shanghai, China) 1,4-Dioxane, mesitylene, tetrahydrofuran (THF), acetone, formic acid, acetic acid, and methanol were obtained from Tianjin Guangcheng Chemical Reagent Co., Ltd. (Tianjin, China). Ethyl acetate, anhydrous ethanol, hydrochloric acid (37%), trifluoroacetic acid, Wahaha pure water, and acetonitrile were obtained from Tianjin Fuyu Chemical Reagent Co., Ltd. (Tianjin, China).

Standard solutions were prepared by dissolving CAP, TAP, or FF (1.000 g) in acetonitrile (100 mL) to obtain a final concentration of 10 mg/mL. The working standard solution was obtained by diluting the stock solution with acetonitrile to the required concentrations for methodological investigation. The above solutions were stored in the fridge under 4 °C.

### 3.2. Instruments and Chromatographic Conditions

FT IR spectra were acquired using a Bruker FT IR spectrometer (Berlin, Germany). The morphology of the 3D COP was imaged using a transmission electron microscope (Hitachi, HT7700, Tokyo, Japan), an 80 kV accelerating voltage and scanning electron microscopy (Hitachi, SU8010, Tokyo, Japan), and a 5 kV accelerating voltage. The PXRD data were obtained using a Rigaku D/Max 2500 PC diffractometer (Tokyo, Japan) with Cu Kα radiation. TGA was conducted with a heating rate of 10 °C·min^−1^ and under nitrogen flow on a TA Instrument Q5 analyzer. BET data were collected by a Micromeritics ASAP 2000 sorption/desorption analyzer (Norcross, GA, USA) using N_2_ adsorption at 77 K. The solid-phase extractions were performed using an extraction instrument (SPR-SPE12), and samples were analyzed by the HPLC (Agilent 1260, Palo Alto, CA, USA), which was equipped with a diode-array detector (DAD) for sample analysis using a detection wavelength of 224 nm and a C_18_ reversed-phase chromatographic column (5 μm, 250 mm × 4.6 mm). Eluent A (acetonitrile) and eluent B (water) (20% acetonitrile in water) were used as the mobile phase to separate CAP, TAP, and FF using isometric elution with a 1.0 mL/min flow speed. The running time for HPLC was 20 min, and the volume of injection was 20 μL.

### 3.3. Synthesis of Imine-Based 3D COP

Tetrakis(4-formylphenyl)-methane (TFPM, 64.8 mg, 0.15 mmol) and 4,4′-diaminobiphenyl (DABP, 55.2 mg, 0.3 mmol) in a mixed solvent of 1,4-dioxane (1.8 mL), mesitylene (0.6 mL), and acetic acid (3 M, 0.3 mL) were put in A Pyrex tube. The above mixture was sonicated for 1 min to obtain the homogeneous dispersion of solids in the solvent. Subsequently, the mixture was sealed, and heated at 120 °C for 3 days. After cooling down to the room temperature, a solid was obtained by filtration, and then purified by Soxhlet extraction in THF by heating to reflux overnight. The solvent was removed under the reduced pressure to afford an imine-based 3D COP (80 mg) as a yellow solid with a yield of 87%.

### 3.4. SPE Procedure

The 3D COP powder was transferred into the SPE cartridge and compacted with a height of 2 cm, which was used as a column. The cartridge was pretreated with acetonitrile and methanol under vacuum until the effluent was colorless to activate the column. The optimized parameter of 3D COP-SPE protocol was listed as follows:

The pH of sample solution was adjusted using hydrochloric acid (0.02 mol/L). Then, the sample in water (20.0 mL) was charged onto the SPE cartridge with a 5.0 mL/min flow speed. The cartridge was dried under vacuum for 3 min. After that, ammonia methanol solution (V_ammonia_/V_methanol_ = 8:92, 2.0 mL) with a flow speed of 3.0 mL/min was used to elute the cartridge. The eluent was completely removed under vacuum and redispersed with methanol (0.05 mL) for HPLC-DAD analysis.

The mixed standard solution (10 mg/mL) was diluted to 3 mg/mL using acetonitrile and then passed through the SPE column filled with 3D COPs at a 0.5 mL/min flow rate. The filtrates were collected by passing through a Millipore filter (0.2 μm) which was obtained from Tianjin Branch billion Lung Experimtental Equipment Co., Ltd. (Tianjin, China) before HPLC analysis.

## 4. Conclusions

An imine-based porous 3D COP was synthesized by solvothermal condensation and was successfully used as a new solid-phase sorbent for SPE. The 3D COP efficiently enriched and detected trace amounts of CAP, TAP, and FF in water samples with ng quantities. The low LODs and LOQs for CAP, TAP, and FF were located in the ng/mL range, indicating the high sensitivity of this method. Hydrophobic effects, π–π interactions, size-matching effects, hydrogen bonding, and excellent chemical stability likely contributed to the excellent enrichment of three analytes in this study. These results indicate that using the 3D COP as a sorbent for SPE can provide an accurate, reliable, and reproducible method to detect trace amounts of three substances often present in environmental water samples.

## Data Availability

Data is contained within the article.

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
