# Peer review of "An Imine-Based Porous 3D Covalent Organic Polymer as a New Sorbent for the Solid-Phase Extraction of Amphenicols from Water Sample"

_molecules, 2023, doi:10.3390/molecules28083301_

Round 1
Reviewer 1 Report
In this work, “An Imine-based 3D Covalent Organic Framework as a New Sorbent for the Solid-phase Extraction of Amphenicols from Water Samples”, Wei and coworkers have reported the application of an organic framework for the solid-phase extraction of amphenicol drugs - chloramphenicol (CAP), thiamphenicol (TAP) and florfenicol (FF) in aqueous solution. The synthesized material is characterized using IR spectroscopy, scanning electron microscopy, transmission electron microscopy, and powder X-ray diffractometry.
The authors have condensed Tetrakis(4-formylphenyl)-methane (TFPM) and 4,4′-diaminobiphenyl (DABP) using an earlier reported method in reference 47, J. Am. Chem. Soc. 2016, 138, 14783−14788. However, their final product does not compare well with the identical 3D-IL-COF-2 reported in the JACS paper. For example, the XRD reported in the Supplementary material significantly lacks the long-range order and the peak at 7.12 degrees differs from the 5.92, 8.49, 12.15, and 13.46° peaks for the earlier reported COF. It should be noted that the term COF is reserved for organic frameworks that possess a 3-dimensional crystallinity which, unfortunately, is missing in this work.
I also did not find the BET surface area of the polymer. It would be helpful for the readers to have an idea of the surface area of the polymer before and after the extraction of drugs.
The extraction part of the experiment is interesting. I suggest the following to make it more effective:
1. Add the structure of the three drugs with the suggested binding sites by H-bonding or pi-pi interaction.
2. Removing the term COF and replacing it with porous polymer or something of that kind. Even though the material reported is not a COF, the focus of this paper is drug detection in ng quantities, which is noteworthy.
The manuscript can be considered for publication after addressing the concerns above.
Reviewer 2 Report
This work reported the solvothermal synthesis of an existing imine-linked 3D COF and its application on the solid-phase extraction and enrichment of CAP, TAP, and FF in water. Necessary factors (e.g., type and volume of eluent, washing speed, pH, and salinity of water) in extraction process were investigated, and the efficiency of COF sorbent was evaluated by correlation coefficient, limit of detection, recyclability etc. However, I do suggest that the authors need to check their results very carefully whether they really obtained the COF product as they claimed before the manuscript acceptance.
1. The author claimed that they applied tetrakis(4-formylphenyl)-methane (TFPM) and 4,4′- diaminobiphenyl (DABP) as the linkers to produce an imine COF reported in the literature (3D-IL-COF-2, Q. Fang et.al, J. Am. Chem. Soc. 2016, 138, 14783−14788). However, according to the literature, the first Bragg reflection peak (2θ) appeared at 5.19°, not 7.12° as the authors showed in Fig. S1. Besides, the quality of the PXRD pattern is too poor to confirm the phase purity, and the data interpretation is confusing. How can only one reflection peak be used to evaluate the phase! The material basis of this study is the synthesized imine COF, if the author could not confirm this basis unambiguously, I am afraid that I cannot recommend an acceptance for its publishing.
2. I do suggest that the authors should re-collect PXRD data for their COF product and simulate a theoretical PXRD pattern according to the reported structure model to see that whether their experimental pattern can match with the simulated one. If not, is it possible that the author obtained a new interpenetration isomer COF since they used different synthesis method and reaction conditions? If so, the author should present the structure model of the new isomer COF with the comparison of the simulated and experimental PXRD patterns.
3. The authors should also 1) reported the calculated BET surface area according to their N2 sorption isotherm, 2) explain why there is a hysteresis, 3) explain why their experimental N2 uptake is lower than the literature result.
4. I suggest that the authors should cite the reference of COF-300 (2009, 131, 4570-4571) which was the first 3D imine COF.
